# Development of a Novel CD26-Targeted Chimeric Antigen Receptor T-Cell Therapy for CD26-Expressing T-Cell Malignancies

**DOI:** 10.3390/cells12162059

**Published:** 2023-08-14

**Authors:** Eiji Kobayashi, Yusuke Kamihara, Miho Arai, Akinori Wada, Shohei Kikuchi, Ryo Hatano, Noriaki Iwao, Takeshi Susukida, Tatsuhiko Ozawa, Yuichi Adachi, Hiroyuki Kishi, Nam H. Dang, Taketo Yamada, Yoshihiro Hayakawa, Chikao Morimoto, Tsutomu Sato

**Affiliations:** 1Department of Immunology, Faculty of Medicine, Academic Assembly, University of Toyama, 2630 Sugitani, Toyama 930-0194, Japan; ekoba@med.u-toyama.ac.jp (E.K.); toz@med.u-toyama.ac.jp (T.O.); immkishi@med.u-toyama.ac.jp (H.K.); 2Department of Hematology, Faculty of Medicine, Academic Assembly, University of Toyama, 2630 Sugitani, Toyama 930-0194, Japan; kamihara@med.u-toyama.ac.jp (Y.K.); akino@med.u-toyama.ac.jp (A.W.); skikuchi@med.u-toyama.ac.jp (S.K.); 3Department of Pediatrics, Faculty of Medicine, Academic Assembly, University of Toyama, 2630 Sugitani, Toyama 930-0194, Japan; mkg8092@yahoo.co.jp (M.A.); ydachi@icloud.com (Y.A.); 4Department of Therapy Development and Innovation for Immune Disorders and Cancers, Graduate School of Medicine, Juntendo University, 2-1-1, Hongo, Bunkyo-ku, Tokyo 113-8421, Japan; rhatano@juntendo.ac.jp (R.H.); morimoto@ims.u-tokyo.ac.jp (C.M.); 5Department of Hematology, Juntendo University Shizuoka Hospital, Izunokuni City, Shizuoka 410-2211, Japan; niwao@juntendo.ac.jp; 6Division of Host Defences, Institute of Natural Medicine, University of Toyama, 2630 Sugitani, Toyama 930-0194, Japan; susukida@inm.u-toyama.ac.jp (T.S.); haya@inm.u-toyama.ac.jp (Y.H.); 7Division of Hematology/Oncology, University of Florida, Gainesville, FL 32610-0275, USA; namdang5@outlook.com; 8Department of Pathology, Saitama Medical University, 38 Morohongo, Moroyama, Saitama 3500495, Japan; taketo@keio.jp

**Keywords:** CD26, CAR-T, T-cell malignancies, third-generation, fratricide

## Abstract

Chimeric-antigen-receptor (CAR) T-cell therapy for CD19-expressing B-cell malignancies is already widely adopted in clinical practice. On the other hand, the development of CAR-T-cell therapy for T-cell malignancies is in its nascent stage. One of the potential targets is CD26, to which we have developed and evaluated the efficacy and safety of the humanized monoclonal antibody YS110. We generated second (CD28) and third (CD28/4-1BB) generation CD26-targeted CAR-T-cells (CD26-2G/3G) using YS110 as the single-chain variable fragment. When co-cultured with CD26-overexpressing target cells, CD26-2G/3G strongly expressed the activation marker CD69 and secreted IFNgamma. In vitro studies targeting the T-cell leukemia cell line HSB2 showed that CD26-2G/3G exhibited significant anti-leukemia effects with the secretion of granzymeB, TNFα, and IL-8, with 3G being superior to 2G. CD26-2G/3G was also highly effective against T-cell lymphoma cells derived from patients. In an in vivo mouse model in which a T-cell lymphoma cell line, KARPAS299, was transplanted subcutaneously, CD26-3G inhibited tumor growth, whereas 2G had no effect. Furthermore, in a systemic dissemination model in which HSB2 was administered intravenously, CD26-3G inhibited tumor growth more potently than 2G, resulting in greater survival benefit. The third-generation CD26-targeted CAR-T-cell therapy may be a promising treatment modality for T-cell malignancies.

## 1. Introduction

CD26 is a 110-kDa surface glycoprotein with intrinsic dipeptidyl peptidase IV (DPPIV) activity that is expressed on various cell types and has many biological functions [1]. An important aspect of CD26 biology is its peptidase activity and its functional and physical association with molecules with key roles in various cellular pathways and biological programs [2]. Immune system CD4 T-cells, especially Th1, Th2, Th17, and T_EM_ (effector memory) cells with high cell-surface CD26 expression, demonstrate anti-tumoral properties [3]. Recent work also suggests that CD26 has significant roles in tumor biology, being both a marker of disease behavior clinically as well as playing an important role in tumor pathogenesis and development [2]. Moreover, we have had a long-standing interest in the role of CD26 in cancer biology and its suitability as a novel therapeutic target in selected neoplasms [4], particularly in view of its expression in such tumors such as malignant pleural mesothelioma (MPM), renal cell carcinoma (RCC), and T-cell malignancies [5].

Regarding MPM and RCC, our previous work showed that cell lines of both tissue types were suitable targets for our mouse anti-CD26 monoclonal antibody 14D10 [6,7]. Based on the amino-acid sequence of 14D10, we produced the recombinant DNA-derived humanized anti-CD26 monoclonal antibody YS110, which is effective against MPM cell lines [7]. Importantly, YS110 exhibited anti-tumor activity in patients with MPM and RCC, as demonstrated in two Phase 1 clinical trials and one Phase 2 trial conducted by our group [8,9,10]. Furthermore, we have shown that CD26 expressed on T-anaplastic large cell lymphoma KARPAS299 cells was involved in cell adhesion and tumorigenicity [11] and that KARPAS299 tumor growth was efficiently suppressed by our anti-CD26 monoclonal antibody 1F7 [12], indicating that CD26 is also a suitable therapeutic target for T-cell malignancies. Given the aggressive nature of T-cell malignancies, the lack of effective treatment and the poor prognosis associated with these cancers [13], there is an urgent need to develop effective therapy for use in the clinical setting. We, therefore, decided to develop a chimeric antigen receptor (CAR)-T-cell therapy based on YS110 given the established efficacy of this treatment modality against B-cell malignancies, even in the refractory setting, to the treatment with tumor-targeting monoclonal antibodies [14].

T-cell aplasia may be a possible on-target, off-tumor toxicity (OTOT) with CAR-T-cell therapy for T-cell malignancies [15,16,17], with acceptance of this side effect likely dependent on severity and treatment efficacy. This scenario may be analogous to how the effectiveness of CAR-T-cell therapy for B-cell malignancies led to acceptance of B-cell aplasia as an OTOT [18].

To establish CAR-T-cells, patient-derived or donor-derived T-cells are transduced with the genetic construct of the CAR to express the single-chain variable fragment (scFv). The scFv is usually designed from efficient antibodies by connecting their variable light (VL) and variable heavy (VH) chain domains with a peptide flexible linker in order to capture a specific antigen expressed on the surface of malignant cells. 

CAR-T-cell therapy targeting CD19 or B-cell maturation antigen (BCMA) on B-cell malignancies is already established as a successful treatment modality in the clinical setting [19,20]. Thus far, only the second-generation of CAR-T-cell therapy targeting these two molecules have been approved, with either CD28 or 4-1BB as costimulatory signals [21]. However, the potential benefits of the third-generation of CD19 CAR-T-cell therapy over the second-generation have been suggested in the setting of clinical trials [22].

Building on the success of this novel approach, we used our well-established anti-CD26 antibody YS110 to develop CD26-targeted CAR-T-cell therapy for T-cell malignancies, demonstrating a greater anti-tumor effect of the third generation compared to the second generation of CAR.

## 2. Materials and Methods

### 2.1. Cell Culture

The T-cell non-Hodgkin’s lymphoma (CD30+ anaplastic large cell lymphoma) cell line KARPAS299 was supplied by the European Collection of Authenticated Cell Cultures (ECACC, Salisbury, UK). The T-lymphoblastic leukemia cell line HSB2, - cell acute lymphocytic leukemia cell line PEER, and T-cell lymphoma cell line (CD30+) Ki-JK were supplied by the Japanese Collection of Research Bioresources Cell Bank (JCRB, Tokyo, Japan). Cutaneous T-cell lymphoma (mycosis fungoides) cell line H9 and T-cell acute lymphocytic leukemia cell line Jurkat were supplied by the American Type Culture Collection (ATCC, Manassas, VA). All cell lines were authenticated by short tandem repeat (STR) profiling and were tested for mycoplasma to be free from contamination at each company. The stable transfectant of Jurkat cells which overexpress CD26 (CD26-Jurkat) has been established as described previously [23,24]. All these cell lines were maintained in RPMI 1640 (Gibco BRL, Tokyo, Japan) supplemented with 10% heat-inactivated fetal bovine serum (Sigma, St. Louis, MO), 100 µg/mL streptomycin, and 100 U/mL penicillin. 

### 2.2. Patient Samples

Among the patients diagnosed with T-cell malignancies between January 2010 and December 2019 at the Division of Hematology, Toyama University Hospital, we randomly picked up biopsy specimens of twenty-one patients which were examined to evaluate the expression of CD26 by immunohistochemistry. The patients’ diagnoses were as follows: anaplastic large cell lymphoma (ALCL) (*n* = 2), angioimmunoblastic T-cell lymphoma (AITL) (*n* = 4), extranodal NK/T-cell lymphoma (NK/T) (*n* = 5), and peripheral T-cell lymphoma (PTCL) (*n* = 10). Lymphoma cells from one PTCL and one AITL patient among the twenty-one patients mentioned above were employed as target cells to evaluate the killing effects of CD26-targeted CAR-T-cells. The PTCL patient exhibited a leukemic phase with 82% lymphoma cell involvement in total white blood cell counts of peripheral blood. The AITL patient had massive infiltration of lymphoma cells in the bone marrow. These lymphoma cells in peripheral blood or bone marrow were separated byi a standard Ficoll-Hypaque technique.

### 2.3. Ethics Approval and Consent to Participate

This study was conducted according to the Declaration of Helsinki and was approved by the ethics committees of Toyama University Hospital (reference number R2019161). Written informed consent was obtained from all patients prior to study participation.

### 2.4. Luciferase-Expression Vector 

The luciferase-expression vector was constructed by the insertion of luciferase gene from the pGL3 Luciferase Reporter Vector Basic (Promega, Madison, WI) into the pMXs-internal ribosomal entry site (IRES)-Kusabira-Orange 2 (KO2), which was prepared by the exchange of green fluorescent protein (GFP) gene in the pMXs-IRES-GFP for the CoralHue^®^ humanized monomeric KO2 gene in the phmKO2-MNL (MBL, Nagoya, Japan).

### 2.5. Flow Cytometry

For flow cytometric analyses, samples were collected using a FACSCanto II flow cytometer (BD Biosciences, San Jose, CA) and analyzed with FlowJo software (Treestar, Ashland, OR). Expression of YS110 scFv was analyzed by its binding with the recombinant human CD26 protein (Fc chimera) (Abcam, Cambridge, MA), which was detected using an APC anti-human IgG F(c) antibody, goat polyclonal (Rockland immunochemicals, Limerick, PA) (Appendix A). Expression of CD8, CD26, human CD45, or mouse CD45 was analyzed using an APC anti-human CD8, clone 3B5, antibody (Invitrogen, Carlsbad, CA), an APC anti-human CD26 antibody, clone BA5b (BioLegend, San Diego, CA), a PE-Cy7 anti-human CD45 antibody, clone 2D1 (BioLegend), or an APC anti-mouse CD45 antibody, clone 30-F11 (Biolegend), respectively.

### 2.6. Immunohistochemical Staining

Expression of CD26 on lymphoma cells in biopsy specimens was assessed by immunohistochemistry using the standard protocol. In brief, following deparaffinization and rehydration, biopsy specimens were incubated with anti-human CD26 mouse monoclonal antibody U16-3 as we previously described [25]. The CD26/anti-CD26 antibody immune complex on the tissue section was detected with the use of the second antibody conjugated with biotin, which was visualized with the 3,3′-diaminobenzidine peroxidase substrate kit (Vector Labs, Burlington, ON). Among the many methods of evaluating and interpreting the immunohistochemical data, we selected the immunoreactive score (IRS) [26], which gives a range of 0–12 as a product of multiplication between (A) positive cells proportion score (0–4) and (B) staining intensity score (0–3). The final IRS score will be a value between 0 and 12; 0–1 = negative, 2–3 = mild, 4–8 = moderate, and 9–12 = strong. 

### 2.7. Construction of CD26 CAR

The basic structures of CD26 CAR-expression vectors were the same as those of TR1 CAR-expression vectors as we previously described [27]. For the anti-CD26 scFv region, we used the VH and VL of our well-established anti-CD26 antibody YS110 [8]. The nucleotide sequences encoding VH (SEQ ID NO: 215, page 85) or VL (SEQ ID NO: 216, page 86) and the amino acid sequences of VH (SEQ ID NO:22; X384) or VL (SEQ ID NO:18; X379) of YS110 (X392 Fab, rhuMAb411) are provided in our patent (US8030469B2; https://patents.google.com/patent/US8030469B2/en accessed on 13 August 2023). The nucleotide fragments of CD26 2/3G and control CD8 2/3G were synthesized (GenScript, Piscataway, NJ) and inserted into the pMXs-IRES-GFP retroviral expression vector RTV-013 (Cell Biolabs, San Diego, CA). As shown in Figure 1A, the second-generation CD26 CAR (CD26 2G/YS110-CD28) contains the leader sequence of the human κ chain (hκ-signal peptide), the YS110 scFv region, the CD8 hinge and transmembrane regions (CD8 hinge-TM), the CD28 intracellular signaling domain (CD28 cyto), and the CD3ζ signaling domain (CD3ζ cyto), in this order. The third -generation CD26 CAR (CD26 3G/YS110-CD28-4-1BB) additionally has the 4-1BB signaling domain (4-1BB cyto) between CD28 cyto and CD3ζ cyto. CD8 controls (CD8 2G/CD8-CD28 and CD8 3G/CD8-CD28-4-1BB) contain the CD8-signal peptide followed by the extracellular domain of human CD8 (CD8) in place of the hκ-signal peptide followed by YS110 scFv of CD26 CAR. The IRES and the GFP genes are linked to the CD26 2/3G and CD8 2/3G as the transduction marker. 

### 2.8. Transduction and Expression of CD26 CAR

Retroviral transduction was performed as we previously described [27]. Briefly, expression vectors of CD26 2/3G and control CD8 2/3G were transfected into Phoenix-A cells (kindly provided by Dr. G. Nolan, Stanford University) using FuGENE^®^6 (Promega). Culture supernatants were collected 72 h following transfection. For retroviral infection into peripheral blood mononuclear cells (PBMCs), retrovirus in the culture supernatants were spin-loaded into 24-well culture plate coated with 20 µg/mL retronectin (TaKaRa, Kyoto, Japan) by centrifugation for 2 h at 2000× *g* at 32 °C according to the manufacturer’s instructions. PBMCs (5 × 10^5^ cells) in 1 mL of medium, which have been stimulated with Dynabeads CD3/CD28 T-Cell Expander (Invitrogen Dynal AS, Oslo, Norway) at a concentration of 25 µL/1 × 10^6^ cells and 30 U/mL of recombinant human intreleukin-2 (rhIL-2) (PeproTech Rocky Hill, NJ) for two days, were added to the retrovirus-loaded 24-well culture plate and spun down at 500× *g* at 22 °C for 10 min, followed by incubation at 37 °C. Following three days of incubation, the cells were transferred into a 25 cm^2^ cell culture flask, and medium containing 30 U/mL of rhIL-2 was added. At day 5 following retroviral transfection, 25 µL of Dynabeads CD3/CD28 T-Cell Expander was added to each cell culture flask.

### 2.9. Activation of CD26 CAR-T-Cells

CD26 2/3G CAR-T-cells and CD8 2/3G control cells were placed onto a 96-well culture plate (1 × 10^5^ cells/well) as the effector cells and were co-cultured overnight with CD26-Jurkat or HSB2 cells as the target cells. For positive controls of the activated effector cells, effector cells were incubated with both 10 ng/mL of phorbol-12-myristate 13-acetate (PMA) (FUJIFILM Wako, Osaka, Japan) and 500 ng/mL of ionomycin (Merck, Darmstadt, Germany) for the same period without the target cells. CD69 expression on effector cells, which were selected from the target cells by gating on marker fluorescence GFP, was then analyzed by flow cytometry using an APC anti-human CD69 antibody, clone FN50 (BioLegend). Furthermore, concentrations of interferon gamma (IFN-γ) and granzyme B in the culture supernatants secreted by the effector cells were measured by the Enzyme-Linked Immuno Sorbent Assay (ELISA) method using a Human IFN-gamma DuoSet ELISA (R&D Systems, Minneapolis, MN) and a Human Granzyme B DuoSet ELISA (R&D Systems). The concentrations of tumor necrosis factor-alpha (TNFα) and inerleukin-8 (IL-8) were also measured by flow cytometry using a BD^TM^ Cytometric Bead Array (CBA) Kit (BD Biosciences) according to the manufacturer’s instructions. 

### 2.10. In Vitro Anti-Tumor Activity of CD26 CAR-T-Cells

HSB2, H9, and KARPAS299 target cells were retrovirally transduced with the luciferase gene by the same method as described above to transduce CD26 2/3G into PBMCs. These target cells were placed onto a 96-well culture plate (1 × 10^4^ cells/well). CD26 2/3G CAR-T-cells or CD8 2/3G control cells were then co-cultured as the effector cells with the target cells for 24 h. The luciferase from the target cells was measured using a Steady-Glo Luciferase Assay System (Promega). For blocking experiments, KARPAS299 target cells were treated with 1 µg/mL of control IgG or YS110 2 h prior to co-culturing. KARPAS299 cells were then co-cultured with CD26 3G CAR-T-cells or CD8 3G control cells for 6 h.

### 2.11. In Vivo Anti-Tumor Activity of CD26 CAR-T-Cells

In vivo experiments were performed as we previously described [28,29]. NOD.Cg-Prkdc^scid^Il2rg^tm1Wjl^/SzJ (NSG) female mice of age (6–7 weeks) and weight (19–21 g) were obtained from Charles River Japan Inc. (Kanagawa, Japan). The mice were kept under specific pathogen-free conditions with a 12 h day and night cycle with free access to food and water, and received humane care in compliance with Institutional Guidelines. All experiments were approved by the animal care and use committee of Toyama University (reference number A2022MED-14). In order to examine the anti-lymphoma activity of CD26 CAR-T-cells, KARPAS299 and HSB2 cells were retrovirally transduced with pMXs luciferase-IRES-KO2 vector mentioned above. Subsequently, 1 × 10^6^ of KARPAS299 cells were transplanted subcutaneously into NSG mice at day 0. 1 × 10^7^ of CD26 2/3G CAR-T-cells or CD8 2/3G control cells were then injected intravenously through the lateral tail veins twice at day 6 and day 13. Tumor volume (mm^3^) was calculated by the formula: ½ (length × width^2^). Luciferase luminescence was estimated using the IVIS Imaging System (PerkinElmer, Waltham, MA) to monitor tumor growth. Signal intensity of the tumor burdens was expressed in Luminescence (photon/sec). To model systemic dissemination, 1 × 10^6^ of HSB2 cells were transplanted intravenously into NSG mice at day 0. 1 × 10^7^ of CD26 2/3G CAR-T-cells or CD8 2/3G control cells were then injected intravenously once at day 1. To monitor tumor growth, bioluminescence imaging using IVIS was performed. Survival of mice was also monitored. In order to identify CD26 3G CAR-T-cells injected into mice, peripheral blood of mice was collected at day 15. Human cells such as CD26 3G CAR-T-cells and HSB2 cells were selected by flow cytometry from live lymphocytes as both human CD45-positive and mouse CD45-negative cells. In human cells, HSB2 cells were identified as the KO2-positive cells and CD26 3G CAR-T-cells were identified as the GFP-positive cells.

### 2.12. In Vitro Anti-Tumor Activity of CD26 CAR-T-Cells against Patient Samples

Lymphoma cells from the patients (1 × 10^5^) mentioned above were co-cultured with CD26 2/3G CAR-T-cells or CD8 2/3G control cells (1 × 10^6^) for 24 h. CD26 2/3G CAR-T-cells and CD8 2/3G control cells were pre-stained with CellTraceTM Oregon Green^®^ 488 Carboxylic Acid Diacetate Succinimidyl Ester (CTOG) (Thermo Fisher Scientific, Ogden, UT, USA) to distinguish these cells from the lymphoma cells by flow cytometry as Alexa Fluor 488-positive cells. The percentage of CD26-positive cells in the Alexa Fluor 488-negative lymphoma cells was a measurement of the anti-tumor effect of CD26 CAR-T-cells. Dead cells were excluded with the staining of Fixable Viability Dye-eFluorTM 780 (eBioscience, Waltham, MA, USA).

### 2.13. Statistical Analysis

All statistical analyses were performed using GraphPad Prism 9 (GraphPad Software, La Jolla, CA, USA). Statistical significance was determined using Student’s t-test. To analyze the statistical significance of differences in survival curves constructed using the Kaplan–Meier method, the log-rank test was used. Statistical significance was defined as *p* < 0.05.

## 3. Results

### 3.1. Transduction and Expression of CD26 CAR

The basic structures of CD26 CAR-expression vectors were the same as those of TR1 CAR-expression vectors as we previously described [27]. For the anti-CD26 scFv region, we used VH and VL of our well-established anti-CD26 antibody YS110 [8] as shown in Figure 1A. Evaluation of GFP fluorescence of PBMCs as the marker of transduction three days following retroviral infection showed a transduction efficacy of approximately 50% for CD26 2/3G and control CD8 2/3G (Figure 1B,C). For these studies, we employed T-cells expressing CD8-signal peptide on the cell surface (CD8 2/3G control cells) as a control for CD26 2/3G CAR-T-cells expressing YS110 scFv. The appropriateness of this control was shown in our previous report [27]. We also examined the expression of YS110 scFv on PBMCs transduced with CD26 2/3G by its binding with the recombinant human CD26 protein (Fc chimera) (Figure 1D). As shown in Figure 1B, almost all of the GFP-positive cells bound to recombinant CD26, as detected by an anti-Fc antibody. CD8 expression on PBMCs transduced with control CD8 2/3G was also confirmed since no CD8-negative cells were detected in the GFP-positive cell population (Figure 1C). As a side note, approximately 30% of cells were CD8-positive in the “no transduction (NT)” PBMCs.

### 3.2. Expansion of CD26 CAR-T-Cells

As shown in Figure 2A, PBMC cell numbers gradually increased following retroviral infection of CD26 2/3G and control CD8 2/3G from day 0 (0.5 × 10^6^ cells) until day 9 (10–20 × 10^6^ cells). We did observe that the number of cells transduced with CD26 2/3G was lower than that of control CD8 2/3G at day 6, likely due to the phenomenon of “fratricide” in the CD26 2/3G CAR-T-cell population [30,31]. Nevertheless, CD26 3G CAR-T-cell level was similar to CD8 3G control cell level by day 9. On the other hand, the CD26 2G CAR-T-cell number was higher than the CD8 2G control cell number at day 9, but then rapidly decreased over the following days. Cells were, therefore, stored on day 9 or 10 and used for subsequent experiments. In order to further analyze the fratricide phenomenon, we assessed CD26 expression at day 3, 6, 9, and 12 (Figure 2B). In contrast to the high level of CD26 expression in the CD8 2/3 control cell population, there was no CD26 expression on CD26 2/3G CAR-T-cells. Our findings were similar to previous work showing loss of CD5 on CD5 2G CAR-T-cells due to fratricide during the expansion process [32].

### 3.3. Activation of CD26 CAR-T-Cells

CD26 2/3G CAR-T-cells and CD8 2/3G control cells were fully stimulated by PMA and ionomycin (P/I), as demonstrated by the high level of CD69 expression and IFN-γ secretion (Appendix A). Meanwhile, co-culture with CD26-Jurkat cells overexpressing CD26 artificially as the target cells (Appendix A) resulted in activation of the CD26 2/3G CAR-T-cell population, as demonstrated by CD69 expression (Appendix A) and IFN-γ secretion (Appendix A). However, such effects were not observed on the CD8 2/3G control cells. Furthermore, the secretion of granzyme B, TNFα, and IL-8 of CD26 3G CAR-T-cells was much higher than that of CD26 2G CAR-T-cells with the statistical significance (Figure 3) when we used a malignant T-cell line, HSB2 cells expressing CD26 originally (Appendix A) as the target cells.

### 3.4. In Vitro Anti-Tumor Activity of CD26 CAR-T-Cells

Then, we examined the anti-tumor effect of CD26 CAR-T-cells on HSB2 cells. CD26 3G CAR-T-cells demonstrated greater anti-tumor activity than CD26 2G CAR-T-cells at both 1: 1 and 5: 1 effector: target cells ratios (*p* < 0.001 and 0.004, respectively) (Figure 4A). As shown in Figure 4B, the anti-tumor activity of CD26 3G CAR-T-cells was further confirmed with the other CD26-expressing T-cell lines (Appendix A), H9 and KARPA299. The anti-tumor effect of CD26 3G CAR-T-cells was completely abrogated by pretreatment of the target KARPAS299 cells with anti-CD26 antibody YS110 prior to co-culture (Appendix A).

### 3.5. In Vivo Anti-Tumor Activity of CD26 CAR-T-Cells

Using a subcutaneous tumor model with the T-lymphoma cell line KARPAS299 (Figure 5A,B), our studies showed that there was no difference in the effect of CD26 2G CAR-T-cells and CD8 2G control cells on tumor growth. In contrast, CD26 3G CAR-T-cells displayed statistically significant anti-lymphoma activity compared with CD8 3G control cells (*p* = 0.019 in Figure 5A and *p* = 0.029 in Figure 5B). We then employed a systemic dissemination model with the T-leukemia cell line HSB2, as described previously [33], with the tumor cells being administered through the tail vein (Figure 5C–E). CD26 2/3G CAR-T-cells significantly inhibited leukemic dissemination compared with CD8 2/3G control cells at day 7 (*p* = 0.000 and *p* = 0.002, respectively) (Figure 5C and Appendix A). The presence of circulating live CD26 3G CAR-T-cells in the mouse peripheral blood was confirmed at day 15 (Appendix A). At day 21, the relative leukemic burden in mice treated with CD26 3G CAR-T-cells was significantly less than mice treated with CD26 2G CAR-T-cells (*p* = 0.026) (Figure 5D). The median survival period of mice treated with CD26 3G CAR-T-cells was 37 days, which was significantly longer than the median of 32 days for mice treated with CD26 2G CAR-T-cells (*p* = 0.002) (Figure 5E).

### 3.6. Anti-Tumor Activity of CD26 CAR-T-Cells against Patient Samples

CD26 expression was detected through immunohistochemical staining on all the biopsy specimens of patients with T-cell malignancies (*n* = 21), which were randomly selected (Table 1). The intensity of staining was mild in five, moderate in nine, and strong in seven samples (typical staining results are demonstrated in Appendix A). Furthermore, lymphoma cells from the peripheral blood of one CD26-positive (moderate) PTCL patient with leukemic phase involvement and from the bone marrow of one CD26-positive (moderate) AITL patient with massive bone marrow involvement were co-cultured as the target cells with CD26 2/3G CAR-T-cells or CD8 2/3G control cells (Figure 6). The level of CD26-positive cells from the PTCL sample were 50-70% following co-culturing with CD8 2/3G control cells while co-culturing with CD26 2/3G CAR-T-cells reduced the level of CD26-positive cells to 1-2% (Figure 6A). Similar results were obtained with the AITL sample (Figure 6B). As supplemental data, CD26 expression across various immature and mature T-cell malignancies are presented in Appendix A. 

## 4. Discussion

In the present study, we have demonstrated that the novel CD26-targeted CAR-T therapy based on our previously developed YS110 antibody may be a promising treatment modality for CD26-expressing T-cell malignancies, particularly third-generation CAR-T-cells containing both CD28 and 4-1BB as co-stimulatory domains.

The CD28 co-stimulatory domain in CD19 2G CAR-T-cells results in the rapid expansion and immediate differentiation into short-lived T effector cells, whereas 4-1BB in CD19 2G CAR-T-cells induces limited T-cell differentiation with the development of central memory T-cells and longer immune control. Despite these differences in biological activity, it is heretofore unclear as to which co-stimulatory domain produces superior clinical outcomes [30]. Of course, the functionality of CAR depends not only on co-stimulation molecules but also the combination of these with scFv, the linker, and the hinge-final 3D configuration. The CD19 CAR-T-cells currently used in the clinical setting are the second-generation products, which have as their co-stimulatory domain either CD28 (axicabtagene ciloleucel; axi-cel) or 4-1BB (tisagenlecleucel; tisa-cel and lisocabtagene maraleucel; liso-cel). However, CD28 and 4-1BB signal through different pathways and may have complementary functions. Indeed, previous studies have demonstrated that CD20 3G (CD28 and 4-1BB) CAR-T-cells exhibited greater activation of intracellular signaling pathways, more potent antitumor activity, and longer in vivo persistence than CD20 2G (CD28) CAR-T-cells [34,35]. The potential superiority of CD19 3G has been suggested in a clinical trial, in which CD19 2G CAR-T-cells (CD28) and CD19 3G CAR-T-cells (CD28 and 4-1BB) were infused simultaneously in 16 patients with relapsed or refractory non-Hodgkin’s lymphoma. This trial demonstrated that CD19 3G CAR-T-cells had superior expansion and greater persistence than CD19 2G CAR-T-cells [22]. However, more work still needs to be done to determine whether CD19 3G is clinically superior to CD19 2G.

The nature of the co-stimulatory domain of CAR is an important issue in the usage of CAR-T-cell therapy in T-cell malignancies, since normal T-cells share most targetable surface antigens with T-cell malignancies. Therefore, CAR-T-cells targeting T-cell malignancies also target each other, resulting in the phenomenon of “fratricide”. This problem was well-described in previous work involving CD5 CAR-T-cells [30]. CD5 2G (4-1BB) CAR-T-cells failed to expand due to fratricide since the TRAF signaling from 4-1BB upregulated the level of intercellular adhesion molecule 1, which stabilized the fratricidal immunologic synapse between CD5 CAR-T-cells [30]. In contrast, the limited fratricide associated with CD5 2G (CD28) CAR-T-cells allowed for sufficient cellular expansion to exert antitumor activity toward T-cell malignancies [32]. Interestingly, the expansion of CD5 CAR-T-cells was accompanied by the downregulation of CD5, possibly facilitating the expansion of CD5 CAR-T-cells and limiting fratricide. Of note is that CD5 downregulation possibly occurred at the translational and/or post-translational level since overall transcription of the CD5 gene was unaltered [32]. This point is particularly important because downregulation of the cognate antigen on tumor cells is a mechanism of escape, and may partly explain the decreased level of target-bearing cells, in addition to direct cell killing effect of CAR-T.

Similar results have been described with YS110-based CD26 2G CAR-T-cells by other researchers [31,36]. Their work showed that CD26 2G (4-1BB) CAR-T-cells exhibited poor viability, multiple cytokine secretion, and direct cytotoxicity against themselves, resulting in fratricide [31]. In contrast, the initial expansion of CD26 2G (CD28) CAR-T-cells were delayed due to transient fratricide, but subsequent expansion of CAR-T-cells was accelerated with the downregulation of CD26. These CD26 2G (CD28) CAR-T-cells displayed cytotoxicity against CD26-positive malignant cells, activated multiple effector functions in co-culture assays, and limited tumor progression in a mouse model [36].

Expansion of both CD26 2G (CD28) CAR-T and CD26 3G (CD28 and 4-1BB) CAR-T-cells associated with CD26 downregulation was also observed in our experiments. Both CD26 2/3G CAR-T-cell populations might undergo transient fratricide at the beginning of culture (Figure 2), consistent with previous reports [32,36]. Similar to CD5, downregulation of CD26 could possibly result from translational or post-translational modifications [32]. Another potential mechanism for decreased CD26 surface expression may involve CD26 intracellular translocation upon YS110 stimulation [37].

In addition, we also observed the accelerated expansion and rapid decrease of CD26 2G CAR-T-cells compared with CD26 3G CAR-T-cells and CD8 2/3G control cells (Figure 2A). This phenomenon may be caused by the stimulatory effects of YS110 scFv itself. We previously reported that crosslinking of CD26 and CD3 with solid-phase immobilized monoclonal antibodies could induce T-cell activation and interleukin-2 production and that anti-CD26 antibody treatment of T-cells enhanced the tyrosine phosphorylation of signaling molecules such as CD3ζ and p56^lck^ [38,39,40,41,42]. YS110 scFv might stimulate CD26 2G CAR-T-cells prior to the downregulation of CD26 in combination with Dynabeads CD3/CD28 and rhIL-2, resulting in the rapid expansion and subsequent exhaustion of the cell population. On the other hand, YS110 scFv-mediated stimulation in the CD26 3G CAR-T-cells might be stabilized by signaling from 4-1BB since 4-1BB activates the noncanonical nuclear factor κB (ncNF-κB) signaling cascade, which is necessary for the survival and persistence of CD19 CAR-T-cells [43]. It is our hypothesis that the lack of ncNF-κB activation could likely result in overstimulation of CD26 2G CAR-T-cells from CD26-mediated signaling.

Most importantly, our present findings demonstrated the successful development of a third-generation CD26-targeted CAR-T with superior in vitro and in vivo activity than the second-generation product against T-cell malignancies. This difference may partly be due to the longer survival of CD26 3G CAR-T-cells compared to CD26 2G CAR-T-cells since our data demonstrated a sharp decline following a significant increase after day 6 of CD26 2G CAR-T-cells (Figure 2A); however, CD26 3G CAR-T-cells were detected in mouse peripheral blood after 15 days post-administration (Appendix A) as well as CD26 2G CAR-T-cells.

Based on previous work, it is our hypothesis that the superior efficacy of CD26 3G CAR-T-cells against T-cell malignancies as compared to CD26 2G CAR-T-cells is at least partially due to 4-1BB-mediated ncNF-κB activation [43]. The 4-1BB/ncNF-κB signaling cascade might affect the degree of CAR-T-cell activation. In our present study, we have clearly demonstrated that the secretion of granzyme B, TNFα, and IL-8 by CD26 3G CAR-T-cells was higher than that seen with CD26 2G CAR-T-cells (Figure 3). This result was compatible with a previous report demonstrating that CD20 3G CAR-T-cells secreted much higher levels of IFN-γ than CD20 2G (CD28) CAR-T-cells [35]. Given the aggressive nature of T-cell malignancies, the lack of effective treatment and the poor prognosis associated with these cancers [13], a novel therapy that is effective can potentially be very beneficial in the clinical setting. While the in vivo survival-prolonging effect of CD26 3G CAR-T-cells may be modest, our data do not preclude the possible use of CD26 3G CAR-T-cell therapy in the clinical setting. Of note is that while CD19 2G (CD28) CAR-T-cell therapy prolonged mouse survival by only one week compared to controls [44], it has demonstrated impressive efficacy in real-world clinical practice, as noted above. While the CD26-targeting CAR-T-cell therapy was highly efficient in the subcutaneous model, it prolonged the survival of mice in the intravenous model for 5 days, which would be equivalent to approximately 6 months for humans. These findings would suggest that the novel therapy may be more appropriate for T-cell lymphoma subtypes rather than for leukemic T-cell malignancies.

The development of CAR-T-cell therapy for T-cell malignancies is underway, with CD5, CD7, and CD30 being targeted in these efforts [21]. In particular, Phase 1 trial of CD7 CAR-T-cell therapy has been completed and excellent response rates have been reported [45]. In this context, a potential advantage of CD26 over CD5, CD7, and CD30 as a target is its expression frequency in T-cell malignancies. CD5, CD7, and CD30 expression levels in PTCL were 85%, 50%, and 16%, respectively; in AITL, 96%, 57%, and 32–50%, respectively; in ALCL, 30%, 32–54%, and 93%, respectively [46]. On the other hand, as shown in Table 1, all 21 patients with PTCL, AITL, ALCL, and NK/T tested expressed CD26, ranging from mild to strong intensity. This ubiquitous expression of CD26 in T-cell malignancies may result in high level of efficacy for CD26 CAR-T-cell therapy in the clinical setting. Of note, our immunostaining assays used the anti-human CD26 mouse monoclonal antibody U16-3, which was developed by our group with high sensitivity and specificity in formalin-fixed, paraffin-embedded samples [25].

A concern for the practical application of CD26 CAR-T-cell therapy in the clinical setting is CD26 expression in normal tissues. Since CD26 is highly expressed on normal T lymphocytes [47], lymphopenia is an expected side effect of CD26 CAR-T-cell therapy. Cordero and colleagues demonstrated that most human CD26-negative CD4 T-cells in circulating lymphocytes are central memory cells while CD26-high expression is present on effector Th1, Th2, Th17, and TEM (effector memory) cells. Therefore, future clinical trials should monitor the levels of these cell types [3]. In addition, since CD26 expression is observed in salivary glands and smooth muscle [47], salivary gland inflammation and gastrointestinal symptoms may potentially be associated with CD26 CAR-T-cell therapy. In the Phase 2 study of YS110, lymphopenia was [3] seen in 45% of patients, while diarrhea and nasopharyngitis were observed in 13% of patients [9]. Furthermore, soluble CD26 level and associated DPPIV activity may be reduced with CD26 CAR-T-cell therapy, findings which were observed with the administration of YS110 [9]. A potential clinical manifestation of this phenomenon may be hypoglycemia, an infrequent side effect associated with administration of DPPIV inhibitors to diabetic patients [48]. Clinical trials are needed to determine the side effect profiles of CD26 CAR-T-cell therapy, particularly since YS110 does not bind to mouse CD26 (unpublished data), hence preventing the assessment of adverse events associated with CD26 CAR-T-cell therapy in a mouse model.

In summary, CD26-targeting CAR-T-cell therapy is a promising therapeutic alternative against T-cell malignancies. Our findings indicate that a third-generation CAR-T product combining both CD28 with 4-1BB co-stimulatory domains displays more anti-tumor activity than the second-generation product harboring CD28 alone. We plan to further validate our conclusion in future studies involving multiple patient donors.

## 5. Conclusions

In this study, we developed and evaluated the efficacy and safety of CD26-targeted CAR-T-cell therapy for T-cell malignancies using YS110 as the single-chain variable fragment. We demonstrated that CD26 3G CAR-T-cells exhibited superior anti-leukemia effects than CD26 2G CAR-T-cells in vitro and in vivo. Our findings suggest that CD26 3G CAR-T-cells are promising candidates for the treatment of T-cell malignancies, which are currently challenging to treat with conventional therapies. However, further studies are needed to optimize the design and delivery of CD26-targeted CAR-T-cells, as well as to assess their long-term safety and efficacy in clinical settings.

## Figures and Tables

**Figure 1 cells-12-02059-f001:**
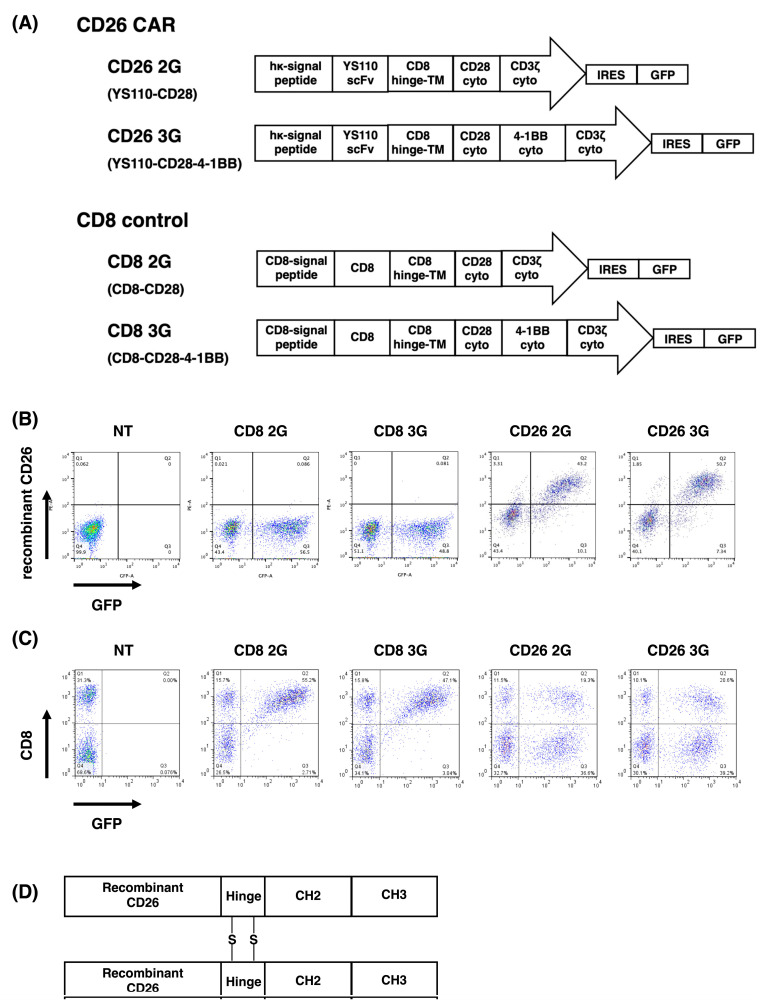
Transduction and expression of CD26 CAR. Gene transduction of CD26 2/3G and control CD8 2/3G into PBMCs was confirmed by GFP marker fluorescence three days following retroviral infection. (**A**) Schematic diagram of CD26 CAR and CD8 control. (**B**) Expression of YS110 scFv on CD26 2/3G-transfected PBMCs was confirmed by binding to Fc-tagged recombinant CD26, which was detected by the anti-Fc antibody conjugated with PE. (**C**) Expression of CD8 on control CD8 2/3G-transfected PBMCs was confirmed with the anti-CD8 antibody conjugated with PE. (**D**) Construct schema for rh-CD26 FC-chimera. Data are representative of three independent experiments (*n* = 3).

**Figure 2 cells-12-02059-f002:**
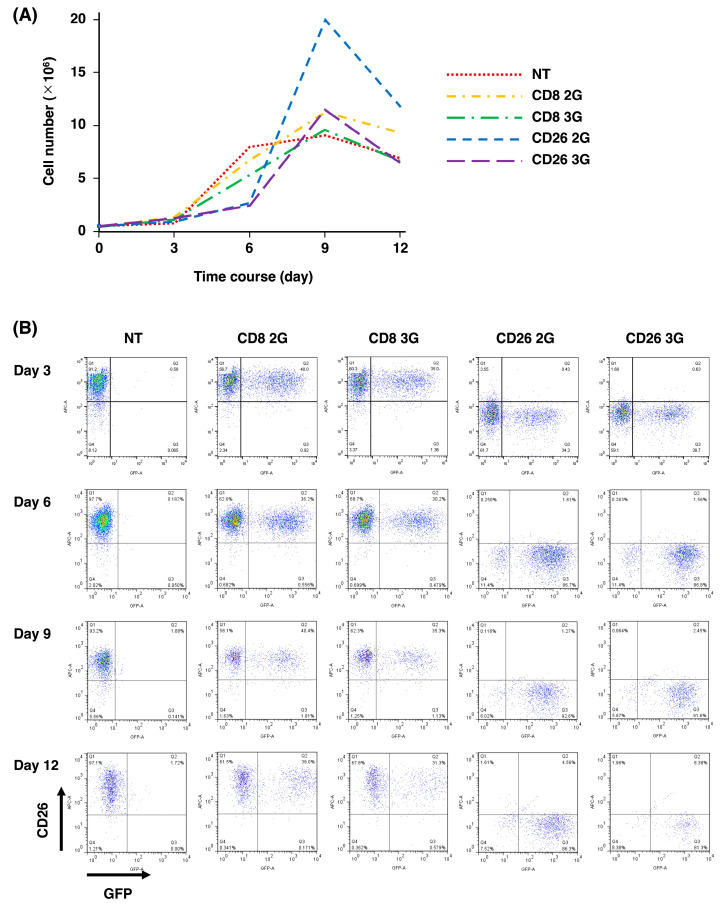
Expansion of CD26 CAR-T-cells. (**A**) Cell numbers of CD26 2/3G CAR-T-cells and CD8 2/3G control cells were determined at day 3, 6, 9 and 12 following retroviral infection. (**B**) CD26 expression on CD26 2/3G CAR-T-cells and CD8 2/3G control cells was analyzed at day 3, 6, 9, and 12 by flow cytometry. Data are representative of three independent experiments (*n* = 1).

**Figure 3 cells-12-02059-f003:**
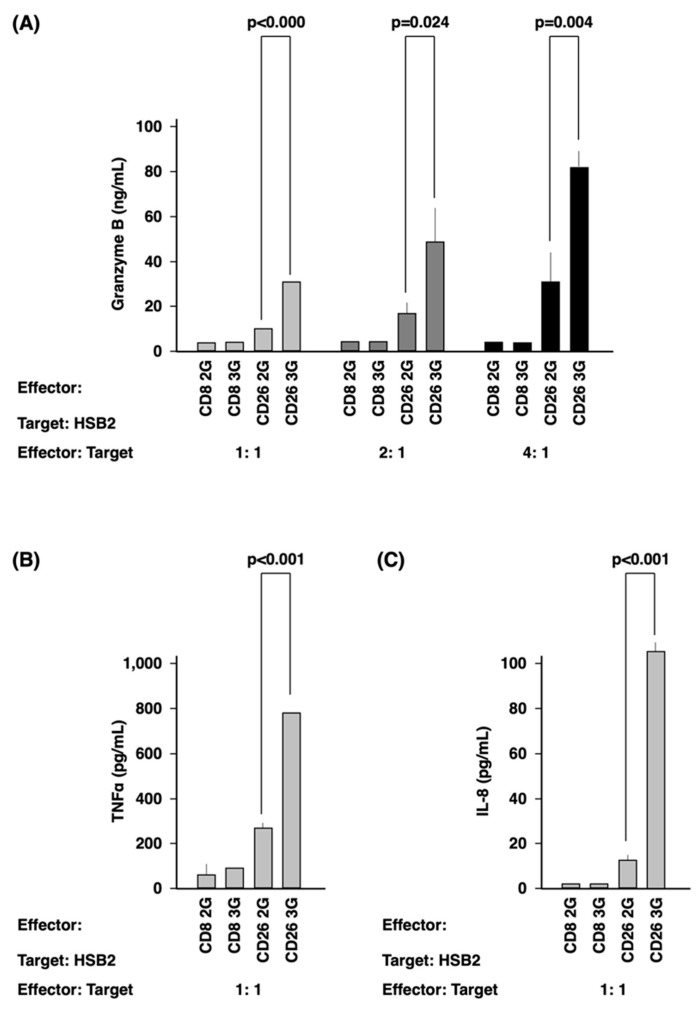
Activation of CD26 CAR-T-cells. Effector cells (CD26 2/3G CAR-T-cells and CD8 2/3G control cells) were cultured with HSB2 cells as target cells at an effector:target ratio of 1:1, 2:1, and 4:1. (**A**) Secretion of granzyme B was evaluated by ELISA assay. (**B**,**C**) Secretion of TNFα (**B**) and IL-8 (**C**) were evaluated by flow cytometry. Data are representative of three independent experiments (*n* = 3).

**Figure 4 cells-12-02059-f004:**
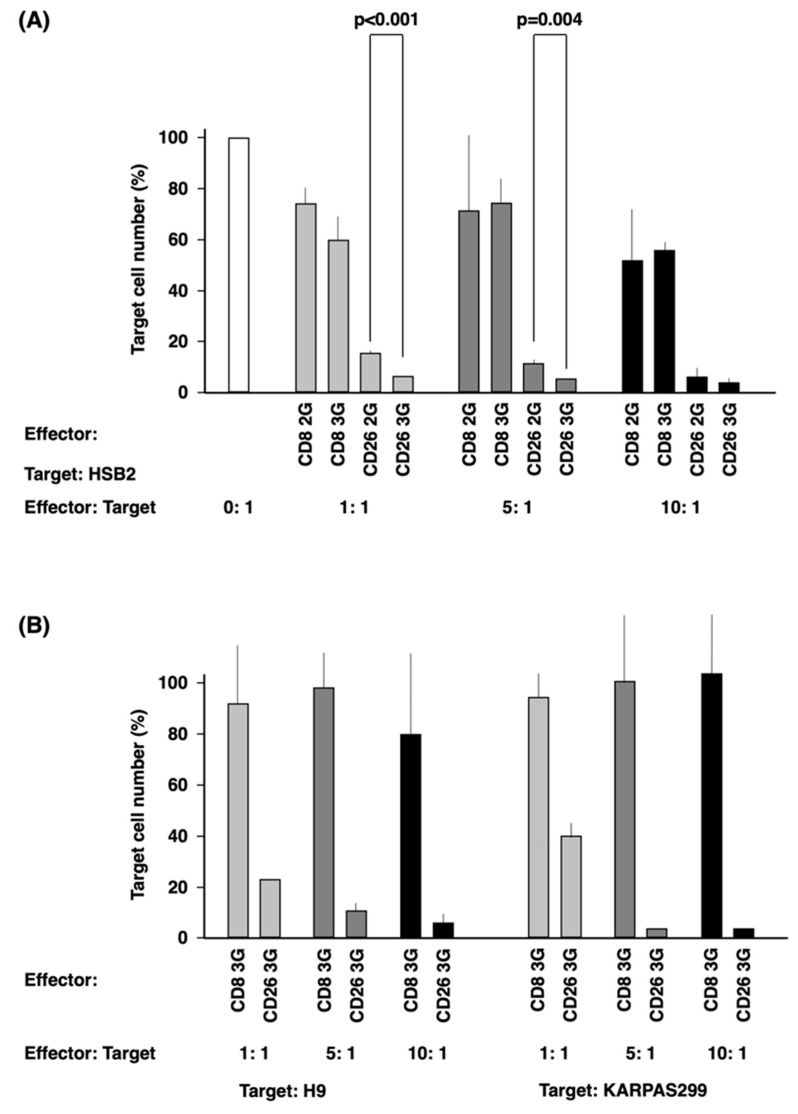
In vitro anti-tumor activity of CD26 CAR-T-cells. (**A**) Target cells (HSB2 cells) were co-cultured with effector cells (CD26 2/3G CAR-T-cells or CD8 2/3G control cells) at an effector:target ratio of 0:1, 1:1, 5:1, and 10:1. (**B**) Target cells (H9 and KARPAS299 cells) were co-cultured with effector cells (CD26 3G CAR-T-cells or CD8 3G control cells) at an effector:target ratio of 1:1, 5:1, and 10:1. Anti-tumor activity of effector cells was evaluated by measurement of luciferase activity from the target cells. Data are representative of three independent experiments (*n* = 3).

**Figure 5 cells-12-02059-f005:**
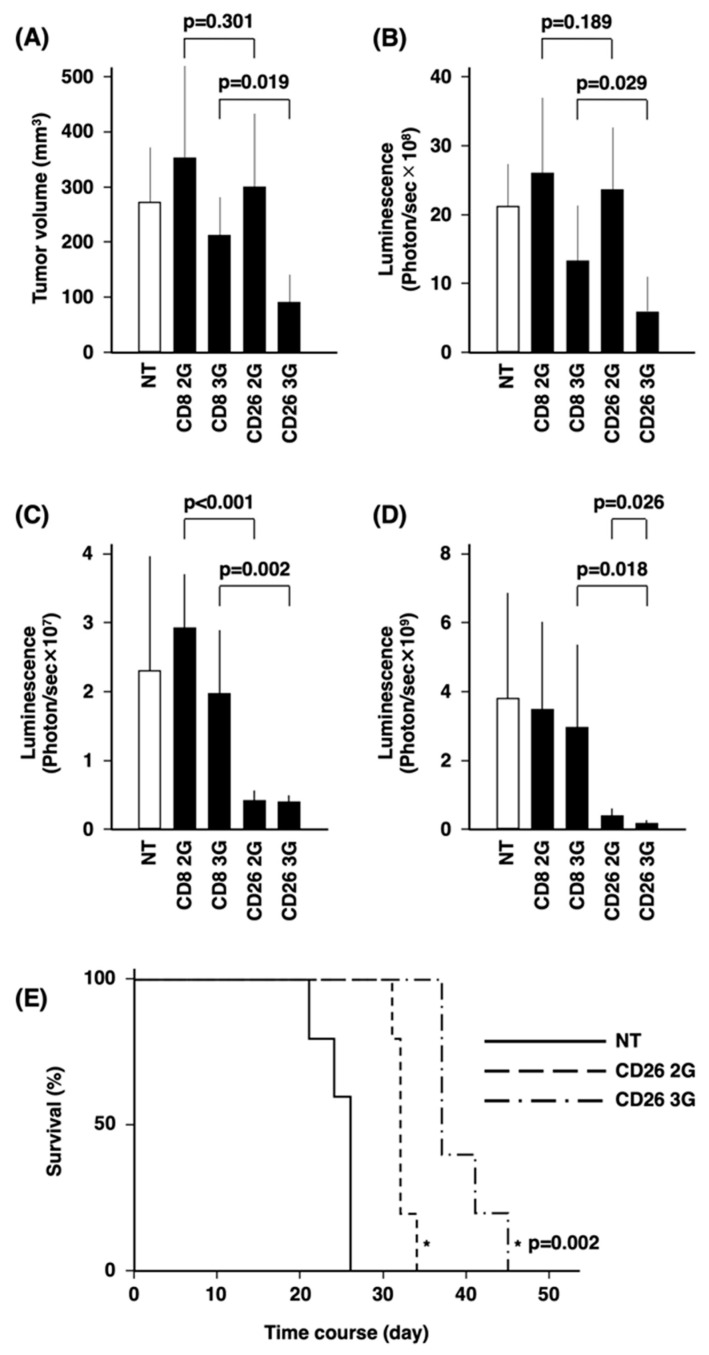
In vivo anti-tumor activity of CD26 CAR-T-cells. (**A**,**B**) KARPAS299 cells transduced with luciferase gene were transplanted subcutaneously into NSG mice at day 0. CD26 2/3G CAR-T-cells or CD8 2/3G control cells were then injected intravenously twice at day 6 and day 13. Tumor volume (mm^3^) (**A**) and luminescence (photon/sec) (**B**) at day 20 are presented (*n* = 6). (**C**–**E**) HSB2 cells transduced with the luciferase gene were transplanted intravenously into NSG mice at day 0 as a murine model of systemic dissemination. CD26 2/3G CAR-T-cells or CD8 2/3G control cells were then injected intravenously once at day 1. Luminescence (photon/sec) at day 7 (**C**) and day 21 (**D**) are presented (*n* = 6). For the same model of systemic dissemination, CD26 2/3G CAR-T-cells were injected intravenously once at day 1 and survival of mice was monitored (*n* = 5) (**E**).

**Figure 6 cells-12-02059-f006:**
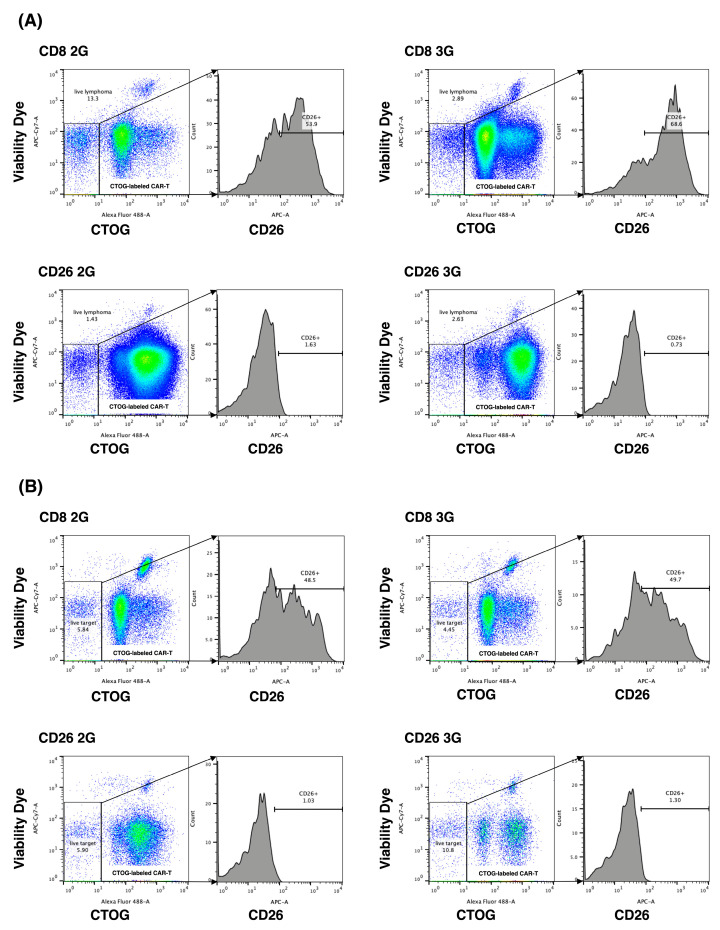
In vitro anti-tumor activity of CD26 CAR-T-cells against patient lymphoma samples. CD26-expressing lymphoma cells from patients with PTCL (**A**) (*n* = 1) or AITL (**B**) (*n* = 1) were co-cultured with CD26 2/3G CAR-T-cells or CD8 2/3G control cells. CD26 2/3G CAR-T-cells or CD8 2/3G control cells were pre-stained with CTOG and were distinguished from lymphoma cells by flow cytometry as the Alexa Fluor 488-positive cells. CD26 expression was evaluated in the Alexa Fluor 488-negative lymphoma cells. Data are representative of three independent experiments (*n* = 3).

**Table 1 cells-12-02059-t001:** The immunoreactive score (IRS) of CD26 in mature T- and NK-cell neoplasms.

	Negative	Positive		
		Mild	Moderate	Strong
ALCL (*n* = 2)	0	0	2	0
AITL (*n* = 4)	0	0	4	0
NK/T (*n* = 5)	0	3	0	2
PTCL (*n* = 10)	0	2	3	5

ALCL, anaplastic large cell lymphoma; AITL, angioimmunoblastic T-cell lymphoma; NK/T, extradodal NK/T-cell lymphoma; and PTCL, peripheral T-cell lymphoma.

## Data Availability

Data are available on reasonable request. All data relevant to the study are included in the article.

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
