# Peer review of "Development of a Novel CD26-Targeted Chimeric Antigen Receptor T-Cell Therapy for CD26-Expressing T-Cell Malignancies"

_cells, 2023, doi:10.3390/cells12162059_

Round 1
Reviewer 1 Report
I apprecdiate the effort of authors. Story, overall quality of data and manuscript are more than sufficient for Cells.
I will recommend accepting your manuscript without further delays. Congratulations! Hopefully your anti-CD26 CAR-Ts will help patients.
Author Response
I appreciate the effort of authors. Story, overall quality of data and manuscript are more than sufficient for Cells. I will recommend accepting your manuscript without further delays. Congratulations! Hopefully your anti-CD26 CAR-Ts will help patients.
We deeply appreciate your very favorable comments.
Reviewer 2 Report
In the current manuscript, Kobayashi, Kamihara and colleagues illustrate the development of a CD26-targeted CAR T therapy for T cell malignancies. The authors have previously evaluated the effect of CD26 CAR T therapy on mesothelioma and renal cell carcinoma and, given the high expression of this marker in T cell malignancies, they propose it as a possible therapeutic strategy in this disease setting.
In the current work, they illustrate the development of a 2nd generation and 3rd generation CD26 CAR T cells and show that the 3rd generation is particularly effective in targeting ALCL, CTCL and T-ALL cell lines in vitro and ALCL lines in vivo. Interestingly, the authors provide a pilot study about the in vitro anti tumor activity of CD26 CAR T cells on primary lymphoma patient samples.
This seems a promising approach to treat T cell malignancies where immunotherapy is so far hampered due to fratricide effects. Also in light of the higher efficacy of 3rd generation constructs in the preclinical setting, the application of this new CAR therapy looks interesting, although possibly more beneficial for patients affected by lymphoma rather than T cell leukemia. Moreover, a huge limitation to the clinical applicability of this approach could come from the high CD26 expression on mature T cells (especially memory) and the cells from the endocrine tissues, GI tract and reproductive system, which may generate strong adverse events, as the authors correctly note in the discussion.
However, the study is noteworthy and well-conducted, although the results are a bit too synthetic. There is a clear style difference between introduction, discussion and results and I feel that, although the scientific data are overall sufficient, the results section is poorly presented and would benefit from extensive editing.
In particular
- Section 3.1: please add more specifications about the CD26 CAR construction. These can be summarized from the methods section. Also, there is no description of Figure 1A in this section.
- Section 3.2: this is a very important point in terms of the fratricide effect of CD26 CAR T cells. Panel 2B shows the percentage of positive cells at day 3 from transduction, but it would be nice to show also data from the following days (6,9,12) to demonstrate the change overtime in the different settings.
- Section 3.3: Figure 3 shows activation markers of CD26 CAR T while co-cultured with malignant cells. Although the data are convincing while using HSB2 cells, it would be nicer if the authors showed also TNF-a, IL-8 and granzyme B levels while using other cell lines having comparable CD26 expression (for example H9) as targets.
- Section 3.5: I advise the authors to show luminescence data in the main figure.
- Section 4: the first paragraph of the discussion is a leftover from the journal's template... Please remove.
Author Response
In the current manuscript, Kobayashi, Kamihara and colleagues illustrate the development of a CD26-targeted CAR T therapy for T cell malignancies. The authors have previously evaluated the effect of CD26 CAR T therapy on mesothelioma and renal cell carcinoma and, given the high expression of this marker in T cell malignancies, they propose it as a possible therapeutic strategy in this disease setting.
In the current work, they illustrate the development of a 2nd generation and 3rd generation CD26 CAR T cells and show that the 3rd generation is particularly effective in targeting ALCL, CTCL and T-ALL cell lines in vitro and ALCL lines in vivo. Interestingly, the authors provide a pilot study about the in vitro antitumor activity of CD26 CAR T cells on primary lymphoma patient samples.
This seems a promising approach to treat T cell malignancies where immunotherapy is so far hampered due to fratricide effects. Also, in light of the higher efficacy of 3rd generation constructs in the preclinical setting, the application of this new CAR therapy looks interesting, although possibly more beneficial for patients affected by lymphoma rather than T cell leukemia. Moreover, a huge limitation to the clinical applicability of this approach could come from the high CD26 expression on mature T cells (especially memory) and the cells from the endocrine tissues, GI tract and reproductive system, which may generate strong adverse events, as the authors correctly note in the discussion.
However, the study is noteworthy and well-conducted, although the results are a bit too synthetic. There is a clear style difference between introduction, discussion and results and I feel that, although the scientific data are overall sufficient, the results section is poorly presented and would benefit from extensive editing.
In particular
- Section 3.1: please add more specifications about the CD26 CAR construction. These can be summarized from the methods section. Also, there is no description of Figure 1A in this section.
Thank you for your comment. We have added a description of Figure 1A to section 3.1.
- Section 3.2: this is a very important point in terms of the fratricide effect of CD26 CAR T cells. Panel 2B shows the percentage of positive cells at day 3 from transduction, but it would be nice to show also data from the following days (6,9,12) to demonstrate the change overtime in the different settings.
We agree with your point. We have added data from day 6, 9, and 12 to Figure 2B.
- Section 3.3: Figure 3 shows activation markers of CD26 CAR T while co-cultured with malignant cells. Although the data are convincing while using HSB2 cells, it would be nicer if the authors showed also TNF-a, IL-8 and granzyme B levels while using other cell lines having comparable CD26 expression (for example H9) as targets.
We appreciate your interest in the activation markers of H9. On the other hand, we have confirmed that CD26 3G is effective against H9 as well (Figure 4B). We would greatly appreciate it if you could kindly accept this data as sufficient.
- Section 3.5: I advise the authors to show luminescence data in the main figure.
We have shown the luminescence data in Figure S5. We would greatly appreciate it if you could kindly accept this data as sufficient.
- Section 4: the first paragraph of the discussion is a leftover from the journal's template... Please remove.
We have removed it as you pointed out.
Reviewer 3 Report
The article is logical and rigorous. The experiment content is sufficient. No obvious language and grammar errors
Author Response
The article is logical and rigorous. The experiment content is sufficient. No obvious language and grammar errors
We deeply appreciate your very favorable comments.
Reviewer 4 Report
CART therapy represent an emerging and promising treatment approach for T lymphoid malignancies. CART therapy had shown significant success in the treatment of certain B cell malignancies, such as acute lymphoblastic leukemia and non-Hodgkin lymphomas. However, its application in T lymphoid malignancies, including T cell acute lymphoblastic leukemia (T-ALL) and peripheral T-cell lymphomas (PTCLs), is still in the early stages of research and clinical trials. The challenges in treating T lymphoid malignancies with CART therapy included finding suitable target antigens specific to T cells, as well as overcoming potential toxicities and resistance mechanisms unique to T cell cancers. CD26, also known as dipeptidyl peptidase 4, is a cell surface protein that is expressed on various cell types, including T cells and plays a role in cell adhesion and signaling and has been found to be overexpressed in certain T cell malignancies, such as T-ALL and some types of PTCLs. Clinical trials exploring anti-CD26 CART therapy and its efficacy in treating T cell malignancies were ongoing. These trials aimed to assess the safety, tolerability, and effectiveness of this novel treatment approach. Anti-CD26 3G CART model proposed in this study showed superior efficacy and longer survival compared to 2G CART. The authors very clearly described the CAR production method and all the in vitro and in vivo tests on mouse models that they performed to demonstrate longer survival, higher antitumor activity.Author Response
CART therapy represents an emerging and promising treatment approach for T lymphoid malignancies. CART therapy had shown significant success in the treatment of certain B cell malignancies, such as acute lymphoblastic leukemia and non-Hodgkin lymphomas. However, its application in T lymphoid malignancies, including T cell acute lymphoblastic leukemia (T-ALL) and peripheral T-cell lymphomas (PTCLs), is still in the early stages of research and clinical trials. The challenges in treating T lymphoid malignancies with CART therapy included finding suitable target antigens specific to T cells, as well as overcoming potential toxicities and resistance mechanisms unique to T cell cancers. CD26, also known as dipeptidyl peptidase 4, is a cell surface protein that is expressed on various cell types, including T cells and plays a role in cell adhesion and signaling and has been found to be overexpressed in certain T cell malignancies, such as T-ALL and some types of PTCLs. Clinical trials exploring anti-CD26 CART therapy and its efficacy in treating T cell malignancies were ongoing. These trials aimed to assess the safety, tolerability, and effectiveness of this novel treatment approach. Anti-CD26 3G CART model proposed in this study showed superior efficacy and longer survival compared to 2G CART. The authors very clearly described the CAR production method and all the in vitro and in vivo tests on mouse models that they performed to demonstrate longer survival, higher antitumor activity.
We deeply appreciate your very favorable comments.